# Validation of Submaximal Step Tests and the 6-Min Walk Test for Predicting Maximal Oxygen Consumption in Young and Healthy Participants

**DOI:** 10.3390/ijerph16234858

**Published:** 2019-12-03

**Authors:** Sung Hyun Hong, Hyuk In Yang, Dong-Il Kim, Tomas I. Gonzales, Soren Brage, Justin Y. Jeon

**Affiliations:** 1Exercise Medicine Center for Diabetes and Cancer Patients, ICONS and Department of Sport Industry Studies, Yonsei University, Seoul 03722, Korea; shhong85@yonsei.ac.kr (S.H.H.); hyukin.yang@gmail.com (H.I.Y.); 2Department of Professional Therapy, Graduate School of Professional Therapy, Gachon University, Seongnam, Gyeonggi-do 1342, Korea; dikim@gachon.ac.kr; 3MRC Epidemiology Unit, Institute of Metabolic Science, Cambridge Biomedical Campus, University of Cambridge, Cambridge CB2 1TN, UK; Tomas.Gonzales@mrc-epid.cam.ac.uk (T.I.G.); soren.brage@mrc-epid.cam.ac.uk (S.B.)

**Keywords:** step test, 6-min walk test, validation, fitness

## Abstract

*Background*: This study aimed to test the validity of three different submaximal tests (i.e., 3-min step test with 20.3-cm step box height (3MST_20_), 3-min step test with 30-cm step box height (3MST_30_), and 6-min walk test (6MWT)) in estimating maximal oxygen consumption (VO_2max_) in young and healthy individuals. *Methods*: The 3MST_20_, 3MST_30_, 6MWT, as well as the cardiopulmonary exercise test (CPET) were performed in 73 participants (37 men and 36 women; mean age: 30.8 ± 9.3 years). All participants visited the clinic three times in a random order for anthropometric measurements, three submaximal tests, and the VO_2max_ test. Multiple linear regression analyses were conducted to construct the VO_2max_ prediction equations for each submaximal test. *Results*: The prediction equations developed based on multiple regression analyses for each submaximal tests were as follows: 3MST_20_: VO_2max_ = 86.0 − 10.9 × sex (male = 1, female = 2) − 0.4 × age − 0.1 × weight − 0.1 × heart rate recovery at 30 s (HRR30s); 3MST_30_: VO_2max_ = 84.5 − 10.2 × sex (male = 1, female = 2) − 0.4 × age − 0.1 × weight − 0.1 × HRR30s; and 6MWT: VO_2max_ = 61.1 − 11.1 × sex (male = 1, female = 2) − 0.4 × age − 0.2 × weight − 0.2 × (distance walked·10^−1^). The estimated VO_2max_ values based on formulated equations were 37.0 ± 7.9, 37.3 ± 7.6, and 36.9 ± 7.9 mL∙kg^−1^∙min^−1^ derived from the 3MST_20_, 3MST_30_, and 6MWT, respectively. These estimated VO_2max_ values were not significantly different from the measured VO_2max_ value, 37.3 mL∙kg^−1^∙min^−1^. The estimated VO_2max_ based on the 3MST_20_, 3MST_30_, and 6MWT results explained 73.4%, 72.2%, and 74.4% of the variances in the measured VO_2max_ (*p* < 0.001), respectively. *Conclusions*: The 3MST_20_, 3MST_30_, and 6MWT were valid in estimating VO_2max_ in relatively young and healthy Asian individuals.

## 1. Introduction

Cardiopulmonary fitness has been used as an index of aerobic fitness for several decades [1]. However, direct measurement of cardiopulmonary fitness using gas analyzers can be costly and sometimes unsafe. Thus, indirect measurement has been considered using a step box [2], cycle ergometer [3], and treadmill [4,5]. Since the use of large equipment such as a cycle ergometer and treadmill could be difficult for field tests, various step tests have been developed and used as a surrogate method to estimate maximal oxygen consumption (VO_2max_) [6].

There are two different types of step tests: Incremental multi-stage and single-stage step tests [4]. In incremental step tests, either the step box height or stepping rate is increased in an incremental manner [7], and various responses of participants during and after exercise are used to estimate VO_2max_ [8,9,10]. Although incremental step tests consider more variables such as heart rate responses, stepping rate or step height reached, and rate of perceived exertion (RPE) which is related to the participants’ aerobic capacity, incremental step tests generally take a longer duration and elicit more physical stress [8,9,10]. Conversely, single-stage step tests mostly take a shorter duration and also only use heart rate recovery to estimate VO_2max_ [7]. Among single-stage step tests, the Tecumseh step test (3-min step test using 20.3-cm step box height (3MST_20_)) [6] and the YMCA 3-min step test (3-min step test using 30-cm step box height (3MST_30_)) [2] have been used widely. Since the 3MST_30_ uses a higher step box height, local muscle fatigue and joint pain could be limiting factors in the elderly and participants with very low fitness levels or poor joint conditions [2,11]. Thus, it would be valuable to understand whether the 3MST_20_ is as good as the 3MST_30_ in estimating VO_2max_.

In addition to step tests, the 6-min walk test (6MWT), developed for patients with respiratory diseases, has been used extensively in the elderly [12] and clinical populations [13,14]. Since then, its validity has been tested in populations of various ages and health status; however, conflicting results have been reported [15,16,17,18]. Furthermore, whether the 6MWT is valid in estimating VO_2max_ in relatively young and healthy populations remains unclear. Although single-stage step tests and the 6MWT have been validated previously, they have not been validated in comparison with each other and against actual measured VO_2max_ in the same participants. Therefore, this study aimed to test the validity of three different submaximal tests (i.e., 3MST_20_, 3MST_30_, and 6MWT) in estimating VO_2max_ in young and healthy individuals.

## 2. Materials and Methods

### 2.1. Study Sample

Seventy-three healthy adults (37 men and 36 women; mean age: 30.8 ± 9.3 years, weight: 68.3 ± 13.9 kg, height: 168.2 ± 10.5 cm) were recruited from the university and Community Service Center at Yonsei University via advertising posters on bulletin boards from March 2013 to March 2014. Participants were screened for cardiopulmonary, orthopedic, and musculoskeletal conditions prior to testing using the Physical Activity Readiness Questionnaire and participant self-reported health history [19]. All participants provided written informed consent after explanation of experimental procedures and possible risks and benefits. The study was approved by the Institutional Review Board (IRB) of Severance Hospital, Yonsei University College of Medicine (IRB number: 4-2013-0345).

### 2.2. Experimental Protocol

The experimental protocol consisted of three clinical visits. During their first visit, anthropometric variables, such as height, weight, waist circumference, blood pressure, resting heart rate (RHR), and VO_2max_ were measured. All participants were randomly categorized into two groups (groups A and B) to avoid order effects. The participants in group A underwent the 3MST_20_ on their second visit and 3MST_30_ on their third visit; those in group B underwent the tests in reverse order. The third visit was scheduled 3 to 7 days after the second visit at the same hour of the day to minimize the effects circadian rhythm of the heart rate. All measurements were performed by the same investigator in a quiet and air-conditioned laboratory (temperature, 18–22 °C; humidity, 40–60%). The participants were asked to maintain their daily living activities as usual. However, they were asked to refrain from strenuous exercise for 24 h, drinking alcohol and caffeine for 4 h, and eating or drinking (except water) for 2 h before the test.

### 2.3. Step Tests

Step tests were conducted using the following standardized procedure. First, participants were fitted with a wearable heart rate monitor (Polar, FT1, Kempele, Finland) and then sat in a chair until a steady-state RHR was achieved (i.e., less than 5 beat-per-minute change in heart rate for one minute. Then, participants continuously stepped onto and off the box 24 times per minute for 3 min while heart rate was recorded every minute. Stepping rate was synchronized to a metronome set at 96 beats per minute and monitored throughout testing. Finally, after 3 min of stepping, participants immediately sat down in a chair while heart rate recovery was monitored for 1 min.

### 2.4. 6MWT

The 6MWT was conducted according to standardized procedures provided by the American Thoracic Society [14]. It was conducted in an indoor corridor, and the course was marked by two green corns placed 30 m apart. Prior to the test, the examiner demonstrated the walking form and provided notifications of foul (both feet off the ground). There was no warm-up or practice beforehand. The participants rested for at least 5 min while sitting on a chair located near the starting position. They were asked to walk at their maximal pace to cover as much ground as possible over a 6-min period. Standardized encouragement (“keep going,” “you are doing well,” and “everything is going fine”) was provided by the examiner during the test; other words were not allowed. The walking distance covered in meters after 6 min was recorded.

### 2.5. Cardiopulmonary Exercise Test (CPET)

CPET to measure VO_2max_ was conducted on a treadmill using a computerized cardiac stress testing system (Cardiac Science, Q-stress TM65, Waukesha, WI, USA). The participants underwent 12-lead electromyography to monitor their heart rate and cardiopulmonary stability and wore a non-rebreathing facemask (Hans Rudolph, Rudolph series 7910, Kansas, MO, USA). Oxygen consumption was continuously measured breath-by-breath using a computerized metabolic measurement system (ParvoMedics, TrueOne 2400, Sandy, UT, USA). Borg rating of perceived exertion (RPE) scale was noted every 2-min. The participants walked for a minute (speed, 1 mph; grade, 0%) prior to the test as warm-up and familiarization to the treadmill. A well-trained investigator followed the modified Bruce protocol. VO_2max_ was considered achieved if any two of the following three criteria were met: (1) Respiratory exchange ratio of ≥1.1; (2) heart rate of >90% of age-predicted maximal heart rate; and (3) RPE of ≥18. The age-predicted maximal heart rate (APMHR) was calculated using the following formula: (208−(0.7×age)) [20]. All participants satisfied at least two of the VO_2max_ criteria and were then included in the analysis. 

### 2.6. Statistical Analysis

The statistical significance level in all tests was set at *p* < 0.05, and all analyses were performed using SPSS 25 for Windows (SPSS Inc., Chicago, IL, US). All data were presented as means ± standard deviations. Correlation analyses were performed to assess the relationship between directly measured VO_2max_, anthropometric measurements, and outcome measures from the step tests and 6MWT. Multiple linear regressions were used to construct the VO_2max_ prediction equation to estimate VO_2max_ using variables gained from the step tests and 6MWT. In the main analysis, we used data from all participants for both model building and validation. In the sensitivity analyses, data-splitting was performed randomly to create three groups. Thereafter, model-building (two-third of the tests) and model-validation data sets (one-third of the tests) were developed. The same process was repeated two more times. Paired t-test was used to assess agreement between the estimated and actual VO_2max_. Scatter plots and Bland–Altman plots were constructed to compare the predicted and directly measured VO_2max_; limits of agreement were set to ± 1.96 standard deviations from the mean.

## 3. Results

The study population was comprised of 37 men and 36 women; participant characteristics are reported in Table 1. Among 73 participants, 64 participants completed all tests: Two step tests, 6MWT, and VO_2max_ test. The 3MST_20_ was not assessed in 6 participants due to conflict of schedule and 6MWT was not assessed in 9 participants due to knee condition or conflict of schedule.

As shown in Table 2, heart rate increased up to 108.2 ± 9.8 bpm (57.7% APMHR) and 125.3 ± 14.5 bpm (67.8% APMHR) at the third minute of the 3MST_20_ in the male and female participants, respectively; conversely, the heart rate increased up to 125.3 ± 10.9 bpm (67.0% APMHR) and 150.0 ± 14.6 bpm (81.1% APMHR) at the third minute of the 3MST_30_, respectively. The measured VO_2max_ value was 42.8 ± 7.3 and 31.6 ± 6.3 mL·kg^−1^·min^−1^ in the male and female participants, respectively.

We performed correlation analyses to determine the association between step test heart rate recovery measures and directly measured VO_2max_ (Appendix A). In both the 3MST_20_ and 3MST_30_, heart rate recovery at 30 s (HRR30s) showed the strongest correlation with the measured VO_2max_ (Appendix A); therefore, it was used for the VO_2max_ estimation model. The HRR30s attributed to the 32.8% of VO_2max_ and 45.8% of VO_2max_ for the 3MST_20_ and 3MST_30_, respectively. When the HRR30s was complemented with sex, age, and weight, the model showed 73.4% and 72.2% of VO_2max_, for 3MST_20_ and 3MST_30_ respectively. The distance walked during the 6MWT attributed to 44.1% of VO_2max_, which increased to 74.4% when sex, age, and weight were added to the model (Table 3).

Thereafter, we used multiple linear regressions to develop the prediction equation for VO_2max_ estimation for each of the submaximal tests.3MST_20_: $VO_{2max}=86.0-10.9\times \;sex\;\left(male=1,\;female=2\right)-0.4\;\times age-0.1\;\times \;weight-0.1\times HRR30s\;$3MST_30_: $VO_{2max}=84.5-10.2\;\times \;sex\;\left(male=1,\;female=2\right)-0.4\times \;age-0.1\;\times \;weight-0.1\;\times \;HRR30s\;$6MWT: $VO_{2max}=61.1-11.1\;\times \;sex\;\left(male=1,\;female=2\right)-0.4\times age-0.2\times \;weight-0.2\;\times \;\left(distance\;walked\cdot 10^{-1}\right)\;$

Using these equations, we calculated VO_2max_ for each participant. The estimated VO_2max_ values were 37.0 ± 7.9, 37.3 ± 7.6, and 36.9 ± 7.9 mL∙kg^−1^∙min^−1^ derived from the 3MST_20_, 3MST_30_, and 6MWT, respectively; the R^2^ values between the estimated and measured VO_2max_ were 0.734, 0.722, and 0.744, respectively (Figure 1).

There were no significant differences between the predicted and directly measured VO_2max_ values. Furthermore, the Bland–Altman plot analysis showed no notable difference in the agreement between the estimated and measured VO_2max_ according to sex in all three submaximal tests (Figure 2).

Since data from same participants were used for building the prediction equation and validation, we performed further sensitivity analyses using three-fold cross validation methods; the participants in two groups were used to develop the prediction equation and participants in one group to validate the estimated results. The same methods were repeated three times (Appendix A), and the results were similar to those in our main analyses. The adjusted R^2^ value between the estimated VO_2max_ using the three different estimation equations from the 3MST_20_ and measured VO_2max_ ranged from 0.702 to 0.773. The adjusted R^2^ value between the estimated VO_2max_ using the three different estimation equations from the 3MST_30_ and measured VO_2max_ ranged from 0.661 to 0.803. The adjusted R^2^ value between the estimated VO_2max_ using the three different estimation equations from the 6MWT and measured VO_2max_ ranged from 0.700 to 0.820.

## 4. Discussion

As cardiopulmonary fitness is often used to predict health status, mortality and the prevalence or incidence of diseases [1], safe, convenient, and valid measurement to assess such is of great interest for epidemiological studies. In our study, we have demonstrated that two steps tests (3MST_20_, 3MST_30_) as well as the 6MWT were valid method in estimating VO_2max_ in relatively young and healthy population. We also confirmed that the 3MST_20_ was not inferior to the 3MST_30_ in estimating VO_2max_, suggesting that a lower step height is as good as a higher step height.

Owing to its convenience, many step test protocols have been developed and validated [2,7,10,21]. In incremental step tests, participants’ ability to perform incremental work and physiological response to the work are usually observed [4]; the step height, test duration, stepping rate, RPE, and heart rate response during the test and/or recovery are also considered. However, these tests have several assumptions: Participants’ will to complete the tests, validity and reliability of the RPE, and validity of the APMHR. Thus, the step test protocol, which eliminates these assumptions from the test, might be more valid. In this context, we validated relatively short (3 min) and single-stage (fixed step height and stepping rate) step tests. As the step height significantly influences the participants’ ability to complete the test owing to possible presence of local muscle fatigue, lower extremity joint pain, and physical function impairment, we tested two different 3-min step test protocols at 20.3 and 30 cm, conducted on separate days in random order. As expected, the heart rate increase was higher during the step test using higher step height was used. However, we found no difference in the validity of the two different protocols in estimating VO_2max_, suggesting that the 3MST_20_ is as good as the 3MST_30_ in estimating VO_2max_.

Demonstrating the 3MST_20_ is not inferior to the 3MST_30_ is of interest for several reasons. Owing to the lower step height, the 3MST_20_ is more feasible for those with lower extremity joint problems, obese and unfit individuals. Previously, Bohannon et al. [11] compared between the 6MWT and YMCA step test and found relatively low completion rates in the latter; 51 out of 189 participants did not complete the test. The average age of those who did not complete the step test was 70.4 ± 14.2 years, while that of those who completed the test was 39.9 ± 19.4 years. Similarly, Beutner et al. [2] also reported that 17% of their participants did not complete 3 min YMCA step test; participants who were not able to complete the 3 min tests were older (53.2 vs. 69.3 years) and more obese (BMI: 24.5 vs. 29.5 kg/m^2^). As most study populations had a wide range of age and health status, we can speculate that the 3MST_20_, which uses a lower step height, would be more feasible. Furthermore, our study showed that the 3MST_20_ and 3MST_30_ increased the heart rate up to 62.8% and 73.9% of the APMHR, respectively. In the female participants, the heart rate increased up to 81.1% of their APMHR during the 3MST_30_, which suggests that the 3MST_30_ is a moderate to vigorous intensity exercise. Therefore, the 3MST_20_ can be more suitable for those with potential risks of cardiac events. However, we did not validate the 3MST_20_ for prediction of VO_2max_ in clinical populations in our study.

One of the strengths of single-stage step tests is the lack of influence by the will of participants, especially with shorter durations (e.g., 3 min) and lower step heights. The heart rate only increased up to 60–70% of the APMHR during the 3MST_20_, which suggests relatively low physical stress to the participants during the test. As participants cannot manipulate their own heart rate, the 3MST_20_ using heart rate recovery after submaximal exercise may yield objective results. In this context, single-stage step tests with lower step heights rather than multi-stage step tests may be more reliable. For example, the Chester step test lasts up to 10 min and uses a 30-cm step box with an incremental stepping rate [4,7]; it ends when participants (1) cannot continue the test, (2) have reached 80% of their APMHR, and (3) have reached 10 min into the test. Therefore, single-stage 3-min step tests with low step heights, such as the Tecumseh step test, might be a safe, objective, inexpensive, convenient, and valid test to assess the level of aerobic fitness.

Although both the 3MST_20_ and 3MST_30_ were valid in estimating VO_2max_, it is important to note that the prediction equations also included other variables, such as age, sex, and body weight. When only HRR30s during the 3MST_20_ and 3MST_30_ was used as a predicting variable, a stronger correlation was observed between the HRR30s and actual VO_2max_ during 3MST_30_. When the prediction equation was supplemented with age, sex, and body weight, the prediction equation developed for both step tests explained the actual VO_2max_ to an equal extent.

Our study also validated the 6MWT in estimating VO_2max_. In 1968, Cooper developed and validated a 12-min run test to estimate VO_2max_ in male US Air Force officers and personnel [22]. Later, McGavin et al. [23] modified this test as a 12-min walk test to assess lung function among patients with chronic bronchitis, which later became the 6-min walk test (3MWT) [24]. Since then, the 6MWT has been validated in various clinical populations [25], including those with cancer [26,27], although a recent study did not recommend the use of the 6MWT to assess VO_2max_ in patients with cancer [28]. In the current study, the 6MWT was also valid in estimating VO_2max_ of the relatively young and healthy participants. The VO_2max_ estimated using the prediction equation derived from the 6MWT agrees with the actual VO_2max_ (36.8 vs. 36.9 mL·kg^−1^·min^−1^). When the model-building and model-validation sets were randomly assigned, the estimated VO_2max_ in the 6MWT was still valid, with a correlation coefficient between 0.771 and 0.910.

Our study has strengths and limitations. One of the strengths of our study is that three different submaximal tests were validated in estimating VO_2max_ against the measured VO_2max_ among the same participants. Furthermore, we provided evidence that the use of short-duration step tests using a low step box (20.3 cm), which elicited a significantly lower heart rate than the step test using a high step box, is equally valid in estimating VO_2max_ in young and healthy populations. Traditionally, a good step test encourages higher steps, higher stepping rates, and longer durations; which mimics the maximal test. However, we clearly demonstrated that submaximal tests using heart rate or distance walked in 6 min were sufficient in estimating VO_2max_ in healthy populations. Conversely, our study limitation is the relatively small convenience sample from a university and community service center in Korea; therefore, the participants were generally young and healthy Koreans. Thus, caution should be taken in applying our results to the elderly, clinical populations, and other ethnicities. Another limitation is the use of the same sample for model building and validation in estimating VO_2max_. To overcome this limitation, we performed sensitivity analyses by randomly splitting our sample into three groups and cross-validating them. These analyses showed that our submaximal tests were all still valid in estimating VO_2max_. Further research with a broader range of populations will, however, be required to enhance the validity of the prediction equations of the tests.

## 5. Conclusions

We validated the 3-min step test using two different step box heights and the 6MWT in a relatively young and healthy Asian population; these tests can be used to provide valid estimates of VO_2max_ in epidemiological studies.

## Figures and Tables

**Figure 1 ijerph-16-04858-f001:**
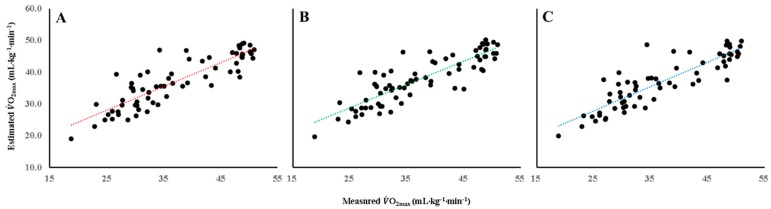
Correlations between measured VO_2max_ and model-predicted VO_2max_ using 2MST_20_ (r = 0.866 *p* < 0.05) (**A**), 3MST_30_ (r = 0.859, *p* < 0.05) (**B**), and 6MWT (r = 0.872, *p* < 0.05) (**C**).

**Figure 2 ijerph-16-04858-f002:**
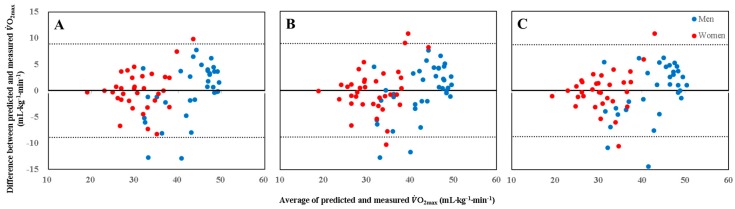
Bland–Altman plots show agreement of mean differences between measured VO_2max_ and estimated VO_2max_ obtained from 3MWT_20_ (**A**), 3MWT_30_ (**B**), and 6MWT (**C**). Solid lines represent mean differences between measured VO_2max_ and estimated VO_2max._ Upper and lower dot lines represent 95% limits of agreement (mean difference ± SD of differences).

**Table 1 ijerph-16-04858-t001:** Characteristics of participants.

Variables	All	Men	Women
*N*	73	37	36
Age (year)	30.8 (9.3)	29.4 (9.5)	32.3 (8.9)
Weight (kg)	68.3 (13.9)	76.7 (11.5)	59.7 (10.7) *
Height (cm)	168.2 (10.5)	175.9 (7.7)	160.3 (6.2) *
BMI (kg/m^2^)	24.2 (3.7)	24.7 (2.7)	23.6 (4.5)
Waist circumference (cm)	81.8 (11.1)	85.4 (9.1)	78.0 (11.8) *
Resting heart rate (bpm)	66.2 (9.4)	63.2 (10.2)	69.2 (7.6) *
SBP (mmHg)	118.7 (13.4)	127.0 (11.9)	110.1 (8.8) *
DBP (mmHg)	75.2 (9.4)	77.0 (9.1)	73.3 (9.6)

Note: BMI = body mass index; bpm = beats per minute; SBP = systolic blood pressure; DBP = diastolic blood pressure. Data are presented as means (standard deviations). * Significant differences from men, *p* < 0.05.

**Table 2 ijerph-16-04858-t002:** The results of maximal and submaximal tests.

Heading	All	Men	Women
3MST_20_	*N* = 66	*N* = 33	*N* = 33
RHR (bpm)	68.9 (9.7)	66.2 (11.0)	71.6 (7.4) *
HR1 (bpm)	109.5 (12.6)	103.6 (10.1)	115.4 (12.3) *
HR2 (bpm)	114.1 (14.6)	105.8 (9.5)	122.6 (14.3) *
HR3 (bpm)	116.7 (15.0)	108.2 (9.8)	125.3 (14.5) *
HRR30s (bpm)	94.1 (15.1)	86.2 (11.4)	101.9 (14.2) *
HRR1 (bpm)	81.1 (13.9)	74.8 (12.2)	87.4 (12.2) *
%APMHR at the third minute of exercise	62.8 (8.7)	57.7 (5.9)	67.8 (8.2) *
3MST_30_	*N* = 73	*N* = 37	*N* = 36
RHR (bpm)	70.7 (10.7)	67.4 (10.3)	74.1 (10.2) *
HR1 (bpm)	121.6 (13.2)	114.3 (8.6)	129.1 (12.9) *
HR2 (bpm)	132.9 (15.6)	122.3 (10.4)	143.8 (12.2) *
HR3 (bpm)	137.5 (17.8)	125.3 (10.9)	150.0 (14.6) *
HRR30s (bpm)	112.9 (20.5)	101.9 (14.9)	124.3 (19.4) *
HRR1 (bpm)	96.5 (20.4)	86.1 (15.0)	107.1 (19.8) *
%APMHR at the third minute of exercise	73.9 (10.8)	67.0 (7.2)	81.1 (9.0) *
Six-minute walk test	*N* = 64	*N* = 32	*N* = 32
Distance (m)	715 (94.9)	762.7 (97.2)	667.4 (64.6) *
Cardiopulmonary exercise test	*N* = 73	*N* = 37	*N* = 36
RHR (bpm)	19.01 (1.1)	19.0 (1.0)	19.0 (1.2)
HR at end of the test (bpm)	188.8 (10.3)	191.6 (8.7)	185.9 (11.2) *
%APMHR	101.3 (5.5)	102.3 (5.0)	100.3 (5.8)
Cessation stage (range)	5.2 (3–6)	5.7 (5–6)	4.6 (3–6)
VO_2max_ (mL∙kg^−1^∙min^−1^)	37.3 (8.8)	42.8 (7.3)	31.6 (6.3) *

Note: 3MST_20_ = 3-min step test using 20.3-cm step box height; RHR = resting heart rate; bpm = beats per minute; HR1 = heart rate at 1 min; HR2 = heart rate at 2 min; HR3 = heart rate at 3 min; HRR30s = heart rate recovery at 30 s after cessation; HRR1 = heart rate recovery at 1 min after cessation; APMHR = age-predicted maximal heart rate; 3MST_30_ = 3-min step test using 30-cm step box height; HR = heart rate; VO_2max_ = maximal oxygen consumption. Numerical variables are presented as means (standard deviations) and categorical variables as means (ranges). * Significant differences from the male participants, *p* < 0.05.

**Table 3 ijerph-16-04858-t003:** Prediction equation model development using the different submaximal tests.

Tests	Models	Adjusted R^2^	SEE	ΔF	Coefficient Estimates					
					Intercept	HRR30s	Sex	Age	Weight	Distance Walked·10^−1^
3MST_20_	Model 1	0.328	7.4	36.7 *	69.9 *	−12.0 *				
	Model 2	0.503	6.4	33.9 *	68.5 *	−0.2 *	−9.0 *			
	Model 3	0.713	4.9	54.7 *	76.4 *	−0.1 *	−8.0 *	−0.5 *		
	Model 4	0.734	4.7	45.9 *	86.0 *	−0.1 *	−10.9 *	−0.4 *	−0.1 *	
3MST_30_	Model 1	0.458	6.5	61.9 *	70.5 *	−0.3 *				
	Model 2	0.555	5.9	45.9 *	70.3 *	−0.2 *	−6.7 *			
	Model 3	0.699	4.8	56.8 *	73.7 *	−0.1 *	−7.6 *	−0.4 *		
	Model 4	0.722	4.7	47.7 *	84.5 *	−0.1 *	−10.2 *	−0.4 *	−0.1 *	
6MWT	Model 1	0.441	6.8	50.8 *	−9.09					0.6 *
	Model 2	0.576	5.9	43.8 *	17.6 *		−7.8 *			0.4 *
	Model 3	0.699	5.0	49.7 *	45.7 *		−8.2 *	−0.4 *		0.2 *
	Model 4	0.744	4.6	46.8 *	61.1 *		−11.2 *	−0.4 *	−0.2 *	0.2 *

Note: SEE = standard error of estimate; HRR30s = heart rate recovery at 30 s; 3MST_20_ = 3-min step test using 20-cm step height; 3MST_30_ = 3-min step test using 30-cm step height; 6MWT = 6-min walk test. * *p* < 0.05.

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
