# Peer review of "Validation of Submaximal Step Tests and the 6-Min Walk Test for Predicting Maximal Oxygen Consumption in Young and Healthy Participants"

_ijerph, 2019, doi:10.3390/ijerph16234858_

Round 1
Reviewer 1 Report
Very interesting research. The introduction of the article presents a good background for research.
The introduction clearly defines the need for such research and defines the purpose of the work.
The methodology of the experiment has been chosen correctly - I have no comments for it.
The statistical tests were selected accordingly. Bland-Altman charts show no statistical differences between the measured values and estimated based on model models values. Therefore, the V̇O2max models for selected tests can be recognized as correct.
Undoubtedly, a big limitation of the work is the fact that the authors used the results from the same group of people to build models and validations. They solved this problem by carrying out sensitivity analyzes, randomly dividing the subjects into 3 groups, cross-validating the results.
However, the authors set out in detail the strengths and limitations of the work in the Discussion section.
The article is suitable for publication, after taking into account two comments:
- In subsection 2.1 (Study sample) the research group should be defined in detail (age, height, BMI, gender etc.). I can see that this data is in the Results section in the first part of the table. Please describe the group in section 2.1 (move part of the table or provide such information in the text).
- At the end of the discussion, please provide information on the directions/possibilities of further development of the conducted research.
Author Response
We would like to appreciate you for careful and thorough reading of the manuscript and for thoughtful comments and constructive suggestions, which help us to improve the quality of the article. Our responses are as follows.
Point 1: In subsection 2.1 (Study sample) the research group should be defined in detail (age, height, BMI, gender etc.). I can see that this data is in the Results section in the first part of the table. Please describe the group in section 2.1 (move part of the table or provide such information in the text).
Response 1:
We thank for the positive feedback from the reviewer. Firstly, as suggested by the reviewer, we have added “Detail information of the population,” as shown in Line 68~69 of the revised manuscript. With regard to the location of the table, it was clearly instructed that tables should be inserted into the main text close to their first citation. That was the reason why we put Table 1 into the result section.
Point 2: At the end of the discussion, please provide information on the directions/possibilities of further development of the conducted research
Response 2:
We thank for the positive feedback from the reviewer. In terms of information on the directions/possibilities of further research: we described strengths and limitations of our study in the discussion section. One of our biggest limitations was the relatively small and homogeneous sample. Therefore, further research should be done with a wide range of populations to fill that void. We have revised the manuscript as follows:
“Further research with a broader range of populations will, however, be required to enhance the validity of the prediction equations of the tests (Line 280~281).”
Reviewer 2 Report
The main objective of this study was to validate the submaximal step and 6-min walk tests to predict maximal oxygen consumption with young and healthy Asian participants.
Authors reported 73-subject data and supported the results with statistical analysis. The limitations were discussed and the results may benefit to others. The manuscript is well-written, however I would encourage authors to incorporate these minor findings,
Line 63, please re-phrase the first sentence or spell out the number. Currently it starts with a number “73”.
Lines 68-74, would you please state when this data was collected (from …… to ……)
Author Response
We would like to appreciate you for careful and thorough reading of the manuscript and for thoughtful comments and constructive suggestions, which help us to improve the quality of the manuscript. Our responses are as follows.
Point 1: Line 63, please re-phrase the first sentence or spell out the number. Currently it starts with a number “73”.
Response 1: As suggested, we have changed 73 to ‘Seventy-three.’
Point 2: Lines 68-74, would you please state when this data was collected (from …… to ……)
Response 2: As suggested, we have revised our manuscript as follows:
"Seventy-three healthy adults (37 men and 36 women; age: 30.8 ± 9.3 yr, weight: 68.3 ± 13.9 kg, height: 168.2 ± 10.5 cm) were recruited from the University and Community Service Center at Yonsei University via advertising posters on bulletin boards from March 2013 to March 2014.”